# The Nephroprotective Effect of Nitric Oxide during Extracorporeal Circulation: An Experimental Study

**DOI:** 10.3390/biomedicines12061298

**Published:** 2024-06-12

**Authors:** Nikolay O. Kamenshchikov, Yuri K. Podoksenov, Boris N. Kozlov, Leonid N. Maslov, Alexander V. Mukhomedzyanov, Mark A. Tyo, Alexander M. Boiko, Natalya Yu. Margolis, Alla A. Boshchenko, Olga N. Serebryakova, Anna N. Dzyuman, Alexander S. Shirshin, Sergey N. Buranov, Victor D. Selemir

**Affiliations:** 1Cardiology Research Institute, Tomsk National Research Medical Center, Russian Academy of Sciences, 111a Kievskaya St., Tomsk 634012, Russia; uk@cardio-tomsk.ru (Y.K.P.); bnkozlov@yandex.ru (B.N.K.); maslov@cardio-tomsk.ru (L.N.M.); sasha_m91@mail.ru (A.V.M.); marik640213@gmail.com (M.A.T.); boiko.cardio@yandex.ru (A.M.B.); nmargolis@yandex.ru (N.Y.M.); bosh@cardio-tomsk.ru (A.A.B.); 2Department of Morphology and General Pathology, Siberian State Medical University, 2 Moskovsky trakt, Tomsk 634050, Russia; oserebryakovan@gmail.com (O.N.S.); dzyman@mail.ru (A.N.D.); 3Federal State Unitary Enterprise “Russian Federal Nuclear Center—All-Russian Research Institute of Experimental Physics”, 37, Mira Ave., Nizhny Novgorod Region, Sarov 607190, Russia; shirshin@ntc.vniief.ru (A.S.S.); sailshir@yandex.ru (S.N.B.); selemir@vniief.ru (V.D.S.)

**Keywords:** nitric oxide, kidney injury, cardiopulmonary bypass, mitochondrial dysfunction, regulated cell death

## Abstract

This study aims to determine the effectiveness of administering 80 ppm nitric oxide in reducing kidney injury, mitochondrial dysfunction and regulated cell death in kidneys during experimental perfusion. Twenty-four sheep were randomized into four groups: two groups received 80 ppm NO conditioning with 90 min of cardiopulmonary bypass (CPB + NO) or 90 min of CPB and hypothermic circulatory arrest (CPB + CA + NO), while two groups received sham protocols (CPB and CPB + CA). Kidney injury was assessed using laboratory (neutrophil gelatinase-associated lipocalin, an acute kidney injury biomarker) and morphological methods (morphometric histological changes in kidney biopsy specimens). A kidney biopsy was performed 60 min after weaning from mechanical perfusion. NO did not increase the concentrations of inhaled NO_2_ and methemoglobin significantly. The NO-conditioning groups showed less severe kidney injury and mitochondrial dysfunction, with statistical significance in the CPB + NO group and reduced tumor necrosis factor-α expression as a trigger of apoptosis and necroptosis in renal tissue in the CPB + CA + NO group compared to the CPB + CA group. The severity of mitochondrial dysfunction in renal tissue was insignificantly lower in the NO-conditioning groups. We conclude that NO administration is safe and effective at reducing kidney injury, mitochondrial dysfunction and regulated cell death in kidneys during experimental CPB.

## 1. Introduction

Cardiac surgery-associated acute kidney injury (CSA-AKI) is a serious, often late-diagnosed complication that occurs in over 40% of patients, 1–7% of whom require renal replacement therapy (RRT) [1]. CSA-AKI leads to increased morbidity and mortality and is not only a medical but also a socio-economic problem. The effectiveness of non-pharmacological strategies to prevent CSA-AKI is controversial, and pharmacological interventions are limited [2]. It is important to identify a pharmacological agent capable of exerting organ-protective effects in cardiac surgery patients with cardiopulmonary bypass (CPB) [3]. Nitric oxide (NO), which has pleiotropic protective properties, could be such a compound [4]. It has been shown that NO decreases the incidence of CSA-AKI [5,6,7]. However, there is currently no evidence that NO can reduce the need for RRT in patients undergoing cardiac surgery under CPB. This area should be the subject of further expanded clinical research.

The molecular mechanism of the renoprotective effect of nitric oxide is unknown. It was hypothesized that the molecular mechanism of the renoprotective impact of NO is similar to the mechanism of the cardioprotective effect of NO donors and includes the activation of intracellular signaling pathways for the realization of an organ-protective phenotype [8]. The end effector of these events is an increase in mitochondrial tolerance to hypoxia/reoxygenation [9]. Mitochondrial dysfunction promotes acute kidney injury [10]; therefore, mitochondria could be suggested to be involved in the renoprotective effect of NO. Also, the renoprotective effect was proposed to be a consequence of the attenuation of the apoptosis, pyroptosis, and necroptosis of renal cells. Pyroptosis and necroptosis are forms of regulated cell death. In pyroptosis, unlike apoptosis, inflammation is an essential component. A key role in pyroptosis is played by caspase-1, which promotes the formation of specific structures of the inflammasome and gasdermin-D. Necroptosis is triggered by tumor necrosis factor (TNF-α), followed by the activation of receptor-interacting protein kinase 3 (RIPK3). Apoptosis, pyroptosis, and necroptosis are known to play important roles in myocardial ischemia reperfusion injury, so they are likely to be involved in kidney injury as well. We used different modalities of mechanical perfusion—prolonged cardiopulmonary bypass (CPB) and prolonged CPB in combination with hypothermic circulatory arrest (CPB + CA), which may be important for clinical practice and planning further studies—to determine potential target populations of responders.

The objective of this study was to determine whether administering 80 ppm nitric oxide is safe and effective at reducing kidney injury, mitochondrial dysfunction, and regulated cell death in kidneys during experimental cardiopulmonary bypass.

The experimental procedures are presented in Figure 1.

## 2. Materials and Methods

This study was performed in 24 sheep of the Altai breed weighing 30–34 kg. All invasive procedures were performed under general anesthesia. All procedures were governed by the Directive 2010/63/EU of the European Parliament and the ”Guide for the Care and Use of Laboratory Animals: Eighth Edition National Research Council”, 2011, by Janet C. Garber. All research protocols were approved by the Ethics Committee of the Cardiology Research Institute, Tomsk NRMC, Protocol No. 230, on 28 June 2022.

### 2.1. Experimental Setting and Animal Preparation

The data of the 24 experimental animals were presented and analyzed, and they were divided into 4 equal groups with 6 sheep in each group as follows: a group receiving CPB according to the standard protocol; a group receiving CPB according to the standard protocol with NO conditioning at a dose of 80 ppm; a group receiving CPB and CA according to the standard protocol, and a group receiving CPB and CA according to the standard protocol with NO conditioning at a dose of 80 ppm. A flowchart of the study design and a flow diagram illustrating the stages of the experiment are shown in Figure 2A,B.

The main phase of the experiment included tracheal intubation with mechanical ventilation and the simulation of CPB or CPB in combination with hypothermic CA. In the “CPB + NO” and “CPB” groups, CPB was carried out under normothermia with a perfusion index of 2 L/min/m^2^ and the maintenance of renal blood flow. In the “CPB + CA + NO” and “CPB + CA” groups, after reaching the target esophageal temperature of 30 °C, the descending aorta was occluded, thus simulating a non-perfusion circulatory arrest for 15 min in which there was no renal blood flow. The perfusion index was lowered to 1 L/min/m^2^. Next, the descending aorta was unclamped, and reperfusion and warming to 36.6 °C were performed. At the warming stage, the perfusion index was 2 L/min/m^2^. In all groups, the mean blood pressure was maintained at >65 mmHg. The cumulative duration of CPB in all groups was 90 min, after which the experimental animals were weaned from CPB, and biopsy specimens were obtained 1 h after spontaneous circulation resumption. Each animal underwent a bilateral biopsy of the anterolateral, upper, and lower pole segments of both kidneys and was euthanized thereafter. A detailed sequence of events, the study methodology, and anesthesia and CPB techniques are available in the Appendix A.

### 2.2. Nitric Oxide Conditioning

For nitric oxide delivery, a device for the plasma–chemical synthesis of nitric oxide, TIANOX, which delivers inhaled NO and monitors NO/NO_2_ in the gas–air mixture supply line directly during NO therapy, was used. In the CPB + NO and CPB + CA + NO groups, the final inspiratory NO concentration was 80 ppm. The animals in the control groups received a standard oxygen–air mixture that did not contain NO. In the group of animals with CPB + NO, nitric oxide was delivered at a dose of 80 ppm immediately after tracheal intubation through the circuit of the ventilator throughout the entire CPB period (90 min) through the circuit of the gas–air mixture of the CPB machine, and within an hour after weaning from CPB through the circuit of the ventilator.

In the group of animals with CPB + CA + NO, nitric oxide was delivered at a dose of 80 ppm immediately after tracheal intubation through the circuit of the ventilator, throughout the entire period of CPB (90 min) through the circuit of the gas–air mixture of the CPB machine, and within an hour after weaning from CPB through the circuit of the ventilator. At the same time, during the stage of non-perfusion hypothermic circulatory arrest (when the target body temperature (esophageal temperature) of 30 °C was reached), the perfusion index decreased to 1 L/min/m^2^, and the descending aorta was occluded for 15 min, NO delivery was not performed since mechanical perfusion in the splanchnic region, including the kidneys, was not carried out. To deliver NO through the circuit of the ventilator, 2 hydrophobic viral and bacterial filters with a Luer connector were built into the inhalation tube. NO was supplied through the proximal one, and gas was taken through the distal one for the continuous monitoring of NO and NO_2_ levels. The inspiratory and expiratory tubes were connected with a Y-adapter. For nitric oxide delivery, a device for the plasma–chemical synthesis of nitric oxide, TIANOX, which delivers inhaled NO and monitors NO/NO_2_ in the supply tube directly during therapy, was used (Figure 1B). To deliver NO through the gas–air tube of the CPB machine, two 1/4 straight connectors with a Luer Lock port were built into the purge gas tube connected to the oxygenator. NO was supplied through the proximal tube, and gas was taken through the distal one for the continuous monitoring of NO and NO_2_ levels. (Figure 1C).

A detailed description of the device for NO conditioning is available in the Appendix A.

The rationale for the dosing and duration of nitric oxide administration is described below.

When choosing a dose and time of NO exposure, clinicians are to be guided by 2 basic principles:The applied dose of nitric oxide and the time of its exposure should be safe for patients;The applied dose of nitric oxide and the time of its exposure should be sufficient to provide potential protective effects.

Recommendations for dosing and perioperative nitric oxide therapy in cardiac surgery are currently developed only for cases of hypoxemic respiratory failure in newborns and young children during the surgical correction of congenital malformation of the circulatory system. Recommended doses range from 3–5 ppm to 40 ppm for reducing pulmonary vascular resistance and to 50–80 ppm for providing hemodynamic support to the injured right ventricle [11].

The mechanisms of increasing NO bioavailability upon its exogenous administration are protein S-nitrosylation and an increase in the serum concentration of NO-NOx metabolites (nitrates, nitrites, S-nitrosothiol, N-nitrosamine, etc.), which serve as reserve donors of NO in the body [12,13]. The accumulation of these metabolites in organs subjected to ischemia–reperfusion explains the organ-protective effects of exogenous NO [14]. In this regard, it seems reasonable to start NO therapy before the start of CPB. It is important to extend the exposure of NO therapy in the postoperative period to the period of early reperfusion after CPB when the main part of organ damage occurs. In our previous study, we tested a new concept of nitric oxide delivery with a fundamentally new technique (into the CPB circuit) and different endpoints (a decrease in CSA-AKI), suggesting extrapulmonary effects of nitric oxide [5]. We found that the concentration of NO metabolites (nitrates, nitrites, and the total concentration of metabolites—NOx) during the postoperative period was lower than at baseline [5]. This was the reason for us to revise the dose of nitric oxide so that it would be increased and to continue NO therapy in the early postoperative period. Therefore, for this study, we chose a dose of NO equal to 80 ppm as optimal for the maximum implementation of the organ-protective effects. The start of NO inhalation at the start of CPB allows the target concentration of NO donors in blood plasma to be achieved; however, its concentration in target organs suffering from ischemia–reperfusion injury does not reach the level that activates organ-protection pathways [14]. The start of NO delivery immediately after intubation makes it possible to increase exposure to the therapy due to the time required by the preparatory stage of surgical intervention (providing surgical access; providing a cannulation scheme for CPB), which averages from 40 to 60 min depending on the experience and qualifications of the operating surgeon. Thus, it is possible to achieve the target concentration of NO and its metabolites in organs and tissues even before the start of CPB and the formation of ischemia–reperfusion cycles. A detailed description of the rationale for the dosing and duration of nitric oxide administration is available in the Appendix A.

### 2.3. Measured and Computed Variables

To achieve the objectives of this study, a statistical analysis comprising pairwise comparisons of variables between the groups included in the experiment was carried out: standard CPB and CPB + NO, CPB + CA and CPB + CA + NO, CPB and CPB + CA, CPB + NO, and CPB + CA + NO. This approach allowed us to explore the possibilities of using setup samples with different modalities of mechanical perfusion since the physiological conditions of CPB and CPB with circulatory arrest are fundamentally different.

The endpoints of the study are described below.

Mitochondrial dysfunction:The mitochondrial transmembrane potential (Δψ) and the mitochondrial permeability transition pore (mPTP) state, as assessed via the mitochondrial calcium retention capacity (CRC);The concentrations of adenosine triphosphate (ATP) and lactate in the kidney biopsies of the experimental animals.Regulated cell death:The concentration of the apoptosis and necroptosis marker tumor necrosis factor α (TNF-228 α);The concentration of pyroptosis markers—nucleotide-binding oligomerization domain (NOD)-like receptor with a pyrin domain 3 (NLRP3) and gasdermin D (GSDMD);The concentration of the necroptosis marker receptor-interacting protein kinase 3 (RIPK3).Kidney injury:The concentration of neutrophil gelatinase–associated lipocalin (uNGAL);Diuresis;The severity of morphological changes in kidney biopsy specimens.

To assess the safety of the proposed technology, we carried out continuous monitoring of the NO_2_ concentration in the gas–air mixture delivered through the ventilator circuit and the CPB oxygenator. The concentration of methemoglobin (MetHb) and the concentration of the final metabolites of nitric oxide—nitrates and nitrites (NO_2_ total; endogenous NO_2_ (eNO_2_); NO_3_)—in the blood of the experimental animals were monitored as well.

To monitor the adequacy of anesthesia and CPB in the stages of the study, we assessed ECG data, invasive arterial pressure, central venous pressure, temperature, oxygen saturation, indicators of acid–base status and blood gas composition, and levels of hemoglobin, hematocrit, lactate, and electrolytes. Blood samples for biochemical studies characterizing the adequacy of mechanical perfusion were obtained at the following stages: 1—immediately after intubation and before the initiation of CPB; 2—at the initiation of CPB; and 3—60 min after weaning the subject from CPB.

Kidney tissues were subjected to standard sample preparation for histological examination. To quantify acute kidney injury in histological preparations, the areas of the renal corpuscle, the glomeruli, and the proximal tubules and their lumens were calculated, followed by the glomerular–capsular index (the ratio of the glomerular area to the area of the Bowman capsule (GCI)) and the lumen–epithelial index (the ratio of the area of the lumen to the area of the epithelium of the proximal tubules (LEI)).

All methods are described in the Appendix A.

### 2.4. Statistical Analysis

Statistical analysis was performed using Statistica 10 (StatSoft Inc., Tulsa, OK, USA) and Jamovi 1.6.16. The normal distribution of quantitative variables was examined using the Shapiro–Wilk test. If the variables had a normal distribution, they were described by mean values and standard deviations, means ± SDs; otherwise, they were described by medians and interquartile ranges, median (Q1; Q3). Differences in quantitative variables in independent groups 1–4 were analyzed using a one-way analysis of variance (one-way ANOVA) or Student’s *t*-test (Welch’s *t*-test with heterogeneity of variance in groups) for independent groups (independent samples t-test) in case of a normal distribution of the variables in all compared groups or using the Kruskal–Wallis test and Mann–Whitney U test otherwise. If a variable had a normal distribution at all three measurement stages, the identification of statistically significant differences in the mean values of the variable at these stages in groups was carried out using a repeated-measures multivariate MANOVA with Mauchly’s test of sphericity and the Greenhouse–Geisser correction or Huynh–Feldt correction in the absence of sphericity. For a post hoc analysis, Student’s *t*-test or a paired samples (paired *t*-test) test with Bonferroni correction for multiple comparisons was used. In the case that there was no normal distribution of the variable in at least at one of the three stages of measurement, to compare the values of the variable at these stages, the Friedman test was used with post hoc comparisons according to the Wilcoxon signed-rank test and corrected for multiple comparisons. In cases where frequency statistics (*p*-values) did not allow for the rejection of the null hypothesis of the equality of means in the compared groups, we applied the Bayes factor. The significance threshold for testing the hypotheses was *p* = 0.05. Detailed descriptions of the strategy and statistical analysis methods used in this study are available in the Appendix A.

## 3. Results

### 3.1. Safety, Metabolic Parameters, and Perioperative Nitric Oxide Homeostasis

In pairs between the groups of animals, dynamics and intragroup and intergroup differences in the main parameters of metabolism, the adequacy of anesthesia and CPB, the safety of NO conditioning, general homeostasis, and nitric oxide homeostasis in the animal body were analyzed. The maximum NO_2_ concentration in the delivered gas–air mixture was 1.4 ppm. The maximum MetHb concentration was 2.8%. The concentration of the final metabolites of nitric oxide—nitrates and nitrites (NO_2_ total, endogenous NO_2_ (eNO_2_), and NO_3_)—in the blood of the experimental animals naturally increased in the NO groups. The data are presented in Table 1 and Table 2.

### 3.2. The Functional State of Mitochondria

The functional state of mitochondria is presented in Table 3.

In the CPB + NO group, the Δψ was on average 36% significantly higher compared to the CPB group, *p* < 0.01, and the power of inference regarding differences was 84.5%. In the CPB + CA + NO group, the Δψ was on average a little higher than in the CPB + CA group, but the *p*-value was equal to 0.24. However, the more sensitive Bayes factor Log10K = 0.628 > 0.5 and *p*-value = 0.24, corresponding to two pieces of information against the null hypothesis, confirm the existence of small differences between the mean Δψ values in the study groups. In the CPB + NO group, the Δψ was on average 40% significantly higher compared to the CPB + CA + NO group; *p* < 0.01.

In the CPB + NO group, the CRC was 69% significantly higher compared to the CPB group, *p* < 0.001, and the power of inference regarding differences was 96.8%. In the CPB + CA + NO group, the mean CRC value was higher compared to the CPB + CA group, and the differences were not statistically significant; *p*-value = 0.21. The more sensitive Bayes factor log10K = 0.833 > 0.5 and *p*-value = 0.21, corresponding to two pieces of information against the null hypothesis, confirm the existence of small differences between the mean CRC values in the study groups.

In the CPB group, the CRC was on average 66% significantly higher compared to the CPB + CA group, *p* = 0.001. In the CPB + NO group, the CRC was on average 184% significantly higher compared to the CPB + CA + NO group; *p* < 0.001. 

In the CPB + NO group, the ATP concentration was on average 86% significantly higher compared to the CPB group; *p* < 0.001. In the CPB + NO group, the ATP concentration was on average 88% significantly higher compared to the CPB + CA + NO group; *p* < 0.001.

In the CPB + NO group, the lactate concentration was on average lower than in the CPB group, but there were no statistically significant differences in the lactate concentration between the CPB and CPB + NO groups; *p*-value = 0.15. The more sensitive Bayes factor amounted to log10K = 0.879 > 0.5. This is consistent with three pieces of information against the null hypothesis and confirms the existence of small differences between the mean lactate concentration values in the study groups. In the CPB + CA + NO group, the lactate concentration was lower compared to the CPB + CA group, but there were no statistically significant differences; *p*-value = 0.14. The more sensitive Bayes factor log10K = 0.893 > 0.5 and *p*-value = 0.14, corresponding to three pieces of information against the null hypothesis, confirm the existence of a small reduction in the mean lactate concentration value in the CPB + CA + NO group compared to the CPB + CA group.

In the CPB group, the lactate concentration was lower compared to the CPB + CA group, but there were no statistically significant differences; *p*-value = 0.22. The more sensitive Bayes factor amounted to log10K = 0.794 > 0.5. This confirms the existence of small differences between the mean values of lactate concentration in these groups.

### 3.3. Regulated Cell Death

The data on regulated cell death are presented in Table 4.

In the CPB + CA + NO group, the TNF-α concentration was 41.5% lower compared to the CPB + CA group; *p* = 0.03. In the CPB + CA group, the TNF-α concentration was higher on average compared to the CPB group, but the differences were not significant. The more sensitive Bayes factor amounted to log10K = 0.736 > 0.5. It confirms the existence of small differences between the mean values of the TNF-α concentration in these groups.

In the CPB + CA group, the RIPK3 concentration was on average higher compared to the CPB group, but with no statistically significant differences; *p* = 0.23. The more sensitive Bayes factor amounted to log10K = 0.763 > 0.5. It confirms the existence of small differences between the mean values of the RIPK3 concentration in these groups. In the CPB + CA + NO group, the RIPK3 concentration was on average higher compared to the CPB + NO group, but with no statistically significant differences; *p* = 0.21. The more sensitive Bayes factor amounted to log10K = 0.81 > 0.5. It confirms the existence of small differences between the mean values of the RIPK3 concentration in these groups. In the CPB + CA group, the RIPK3 concentration was on average higher compared to the CPB group, but with no statistically significant differences; *p* = 0.23. The more sensitive Bayes factor amounted to log10K = 0.76 > 0.5. It confirms the existence of small differences between the mean values of the RIPK3 concentration in these groups.

In the CPB + CA + NO group, the NLRP3 concentration was on average lower compared to the CPB + NO group, but with no statistically significant differences; *p* = 0.21. The more sensitive Bayes factor amounted to log10K = 0.693 > 0.5. It confirms the existence of small differences between the mean values of the NLRP3 concentration in these groups.

### 3.4. Kidney Protection

The uNGAL concentration 60 min after weaning from CPB was as follows: in the standard CPB group, 1.69 ± 0.751 ng/mL; in the CPB + NO group, 0.62 ± 0.268 ng/mL; in the CPB + CA group, 2.23 ± 0.881 ng/mL; and in the CPB + CA + NO group, 0.67 ± 0.255 ng/mL. The total urine output was as follows: in the CPB group, 753 ± 191.3 mL; in the CPB + NO group, 972 ± 89.3 mL; in the CPB + CA group, 678 ± 199.2 mL; and in the CPB + CA + NO group, 683 ± 201.9 mL. The data are presented in Table 1 and Table 2. In the CPB + NO group, the uNGAL concentration was 70% significantly lower compared to the CPB group; *p* = 0.0001. In the CPB + CA + NO group, the uNGAL concentration was 63% lower compared to the CPB + CA group; *p* = 0.0039. In the CPB + NO group, diuresis (the total urine output) was 29% significantly higher compared to the CPB group; *p* = 0.0297. We did not find a statistically significant difference between the CPB + CA and CPB + CA + NO groups in diuresis. In the CPB group, the uNGAL concentration was 32% significantly lower compared to the CPB + CA group; *p* = 0.0342. In the CPB + NO group, diuresis (the total urine output) was 42% significantly higher compared to the CPB + CA + NO group; *p* = 0.0213. The concentration of uNGAL in the CPB + NO vs. CPB + CA + NO groups did not differ significantly. Diuresis in the CPB vs. CPB + CA groups did not differ significantly.

### 3.5. Histological Changes

The Kidney samples were mainly represented by the cortical substance. In all groups, the same types of changes with varying severities were found. The lumen of the proximal convoluted tubules of the nephrons was dilated, the epithelium looked flattened, and the height of the epithelial cells often decreased (a loss of the brush border). Sometimes, fragmentation of the apical region of nephrocytes was observed; in some animals, multiple granularity and small optical empty vesicles (granular and vacuolar hydropic dystrophy) were observed. There were no changes in the distal convoluted tubules of the nephrons. The collecting ducts were dilated. The renal corpuscles often had enlarged urinary spaces (glomerular edema). There was slight interstitial edema (Figure 3).

The GCI and LEI were assessed.

Morphometric data on kidney injury are presented in Table 5.

In the CPB + NO group, the GCI was on average higher compared to the CPB group; *p* = 0.03. In the CPB + CA + NO group, the GCI was on average higher compared to the CPB + CA group; *p* < 0.0001.

In the CPB + NO group, the LEI was on average lower compared to the CPB group; *p* < 0.01.

In the CPB group, the GCI was on average higher compared to the CPB + CA group; *p* < 0.001. In the CPB + NO group, the GCI was on average higher compared to the CPB + CA + NO group; *p* = 0.049.

In the CPB group, the LEI was on average lower compared to the CPB + CA group; *p* < 0.0001. In the CPB + NO group, the LEI was on average lower compared to the CPB + CA + NO group; *p* < 0.0001.

The morphological changes were of the same types; their severity is presented in the following order: CPB > CPB + NO, CPB + CA > CPB + CA + NO, CPB + CA > CPB, CPB + CA + NO > CPB + NO (Figure 3A–D).

## 4. Discussion

We performed an experimental study that simulated CPB or CA to assess the safety and efficacy of perioperative conditioning with nitric oxide produced using a plasma–chemical synthesis method. Our study revealed new and significant insights.

Our study showed that nitric oxide has nephroprotective properties, as evidenced by a significantly lower concentration of a renal damage marker, uNGAL, in groups with NO administration and a significantly higher total urine output in the CPB + NO group compared to the CPB group. Histological signs of damage to renal tissue were also less pronounced in groups with NO administration. Differences in the GCI indicate an increase in urinary space under the influence of NO. Differences in the LEI indicate the preservation of the tubular epithelium under the influence of NO.

When implementing new methods for NO therapy, careful monitoring of safety is necessary, namely, monitoring toxicity associated with NO_2_ production and the formation of a high methemoglobin concentration. In clinical practice, the maximum permissible level of NO_2_ is 3 ppm in an 8 h time-weighted average, while the maximum level of methemoglobin should be kept to less than 5% of the total hemoglobin concentration [15]. In our study, these variables were in the range of acceptable values, which confirms the safety of the applied plasma–chemical synthesis method. At all stages of our study, in the NO-conditioning groups compared to the groups that received the standard protocol of extracorporeal perfusion, there was a significant increase in the concentration of total nitrates and nitrites; however, these changes were natural, and they were not accompanied by an output of gas or metabolic and hemodynamic profiles beyond the range of acceptable values.

Mitochondria are involved in the activation and implementation of the mechanisms of the danger model under critical conditions. Mitochondrial damage in cardiac surgery leads to various types of cell death, the formation of organ dysfunction, and irreversible multiple-organ failure [16]. Mitochondria can act as signaling stations for the regulation of cytoprotective mechanisms and adaptive responses to damaging factors, and mitochondrial dysfunction affects clinical outcomes, including the development of CSA-AKI, and is considered a potential target for therapeutic interventions [17,18]. The pluripotent properties of NO can trigger protective effects directed at mitochondria through several mechanisms: the effects of pharmacological preconditioning, direct antioxidant and anti-inflammatory effects, by indirectly reducing oxidative damage to renal tubules during the conversion of free oxyhemoglobin to MetHb, and by modulating O_2_ consumption under basal and stimulated conditions. [19,20]. The Δψ, which determines the efficiency of ATP synthesis by mitochondria, as well as the mitochondrial CRC, which distinguishes the state of the mPTP, were measured [21]. The mPTP state determines whether a cell lives or dies. mPTP opening induces apoptotic cell death and reduces ATP synthesis [22,23]. In contrast, mPTP closing prevents apoptotic cell death and promotes ATP synthesis [22,23]. An increase in renal mitochondrial tolerance to the negative impact of CPB suggests that NO could improve ATP synthesis and prevent apoptotic cell death. This is true; it was found that NO conditioning contributes to an increase in the ATP level in renal tissue. It is possible that the signaling mechanism of the NO-induced renoprotective effect triggers the following events: (1) the NO-induced activation of soluble guanylyl cyclase; (2) an increase in the cyclic guanosine monophosphate content in the cell; (3) the stimulation of protein kinase G; (4) mPTP closing, mitochondrial ATP-sensitive K^+^ channel opening, and large-conductance Ca^2+^-activated K^+^ channel opening; and (5) an increase in renal mitochondrial tolerance to the negative impact of CPB. These signaling events occur in cardiomyocytes after soluble guanylyl cyclase stimulation by NO [8,9,24,25]. The main source of lactate in the cell is anaerobic glycolysis [26]. CPB promoted an increase in lactate content in renal tissue. Our study showed little effect of NO on renal lactate levels. Altogether, the data obtained indicate the identity of the cytoprotective mechanisms of cardiac preconditioning (ischemic or pharmacological, including induction by anesthetics) in ischemic–reperfusion injury with the final effector pathways of NO conditioning on the kidneys during extracorporeal perfusion [27]. The development of treatment strategies leading to the modulation of cell death is of great research and practical value [28,29]. According to our data, NO did not inhibit pyroptosis or necroptosis. We propose that NO could inhibit apoptosis because NO promotes mPTP closing. The inhibition of apoptosis is a therapeutic target for reducing CSA-AKI [30,31]. The inactivation of mPTP prevents the manifestation of CSA-AKI and the transition to chronic kidney disease [10]. It was reported that TNF-α can induce both necroptosis and apoptosis. Our findings demonstrate that NO inhibited TNF-α production in renal cells in the CPB + CA + NO group. TNF-α is a predictor of adverse renal outcomes and mortality in cardiac surgery [32]. It was reported that NO can inhibit apoptotic cell death [33]; therefore, it could be proposed that NO inhibits apoptosis in renal tissue. In addition, it should be noted that TNF-α is a proinflammatory cytokine; therefore, a reduction in its production could attenuate the inflammatory process in kidneys in response to CPB [34].

It should be noted that the most pronounced effect of NO in relation to the correction of mitochondrial dysfunction was found in the CPB + NO group, while in the CPB + CA + NO group, there was no effect (in terms of ATP concentration), or it was weakly expressed (Δψ; CRC). At the same time, nitric oxide significantly reduced the concentration of the apoptosis marker TNF- α in the CPB + CA + NO group.

We also found that the studied perfusion options have different effects on the kidneys, and the effects of nitric oxide also differ. When comparing the CPB and CPB + CA groups, we revealed a more pronounced damaging effect of CPB + CA which was manifested by significantly higher values of uNGAL and CRC in the CPB + CA group, as well as small differences between the means of the lactate, TNF-α, and RIPK3 concentrations in these groups. Histological signs of renal tissue damage also indicate less severity in the CPB group compared to the CPB + CA group.

When comparing the CPB + NO and CPB + CA + NO groups, we identified a different degree of effectiveness of NO therapy, which was manifested by significantly higher values of urine output, the Δψ, the CRC, and the ATP concentration in the CPB + NO group, as well as small differences between the means of the RIPK3 and NLRP3 concentrations in these groups. Histological signs of renal tissue damage also indicate less severity in the CPB + NO group compared to the CPB + CA + NO group.

These results indicate that extracorporeal perfusion combined with circulatory arrest is more invasive than a CPB procedure alone. NO therapy had a greater effect on many indicators in the isolated CPB group. At the same time, with regard to the correction of necro- and pyroptosis, nitric oxide showed slightly greater effectiveness in the CPB + CA group. Thus, the different results in the two mechanical perfusion models may reflect different degrees of hypoxic stress and severities of lethal/sublethal cellular injuries. This points to a need for further research on NO therapy with different options for perfusion support during operations.

Our data do not exclude the possibility that NO delivery exerts nephroprotective effects through other mechanisms. Due to hemolysis and endothelial dysfunction, CPB can lead to a deficiency in endogenous NO, which is an important signaling molecule affecting the body metabolism, the kidneys, and cardiovascular system functioning [35]. NO delivery can have a nephroprotective effect due to optimizing renal microcirculation, the transport of electrolytes in the renal tubules, vascular tone, blood pressure, platelet aggregation, and immune cell activation [36,37,38].

Our findings have several translationally significant consequences. In a recent experimental study, NO administered with CPB demonstrated serologic and histologic evidence of renal protection from AKI [39]. The authors demonstrated the histological picture. Additionally, the applied CPB protocol included a long period of intentional hypoperfusion, which limits the translation of the obtained data into clinical practice. This is probably due to the use of mechanical perfusion according to clinical protocols, which reduced the aggressiveness and leveled the expression of gross morphological features of the injury. Our study is the first one to demonstrate the possibility of safely performing perioperative conditioning with plasma–chemically synthesized NO to reduce mitochondrial dysfunction with a clinically relevant CPB model and to modulate the apoptotic signal during CA in kidney cells. The targeted NO modulation of mPTP to prevent CSA-AKI is promising, including when used in perioperative care bundles [40]. CSA-AKI has a variety of phenotypes, with different underlying pathophysiological mechanisms, necessitating a multimodal approach that includes exposure to various therapeutic targets (including pathways involved in hemodynamic and oxygen delivery, inflammation, cellular metabolism, oxidative stress, and apoptosis). For this purpose, the combined use of preventive perioperative strategies (the implementation of the KDIGO (Kidney Disease: Improving Global Outcomes) guidelines, goal-directed perfusion, and targeted oxygen delivery during CPB, etc.) is proposed to prevent organ complications [30]. Adjuvant NO conditioning as a pharmacological intervention and a component of complex perioperative management holds great promise for bringing together the complex puzzle of CSA-AKI. Our results suggest patient populations that could benefit from perioperative NO delivery: patients undergoing CA surgery, high-risk patients, and those with comorbidities associated with endogenous NO deficiency (chronic kidney disease, diabetes mellitus, obesity, and hypertension) [41]. The obtained data will allow for the revision of the modality of perioperative NO use with increases in its dose and duration for the purposes of organ protection in cardiac surgery in order to remove the contradictions of the current study results [6,42,43].

This study has a number of limitations, so the results should be extrapolated with caution. First, this study had a relatively short postoperative follow-up period. Next, with the exception of the diuresis rate, we did not examine other markers of renal function (in particular, natriuresis and urinary creatinine concentration). Finally, a detailed morphological picture of biopsy specimens of the outer renal medulla, which is primary important in hypoxic AKI, was not obtained.

## 5. Conclusions

The use of 80 ppm NO is safe and leads to the formation of the kidney protective effect in various types of experimental mechanical perfusion. Nitric oxide gas induced an increase in renal mitochondria tolerance to the negative impact of CPB and reduced kidney injury, mitochondrial dysfunction, and regulated cell death in the kidneys during experimental cardiopulmonary bypass. The therapeutic potential of NO as an intervention to achieve an organ-protective phenotype can be used as a component of a multimodal strategy for the prevention of CSA-AKI.

## Figures and Tables

**Figure 1 biomedicines-12-01298-f001:**
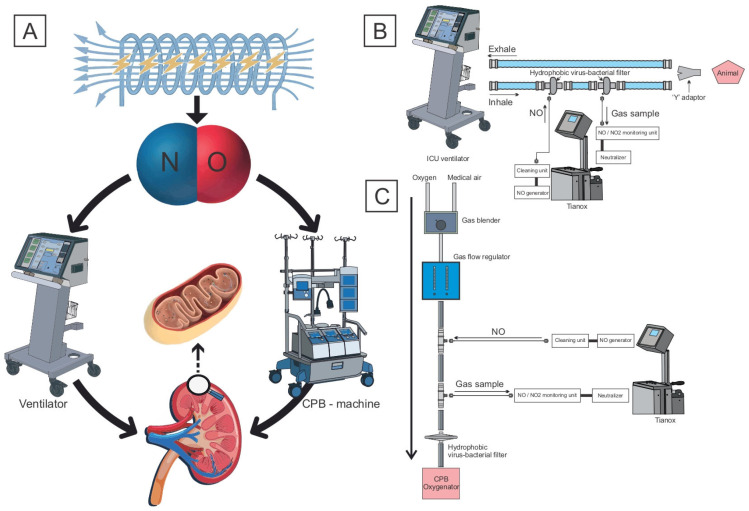
(**A**) Study of the effect of delivering nitric oxide produced by plasma–chemical synthesis on the severity of mitochondrial dysfunction in kidneys under different types of mechanical perfusion. (**B**) NO delivery through the circuit of the ventilator; (**C**) through the gas–air line of the CPB machine.

**Figure 2 biomedicines-12-01298-f002:**
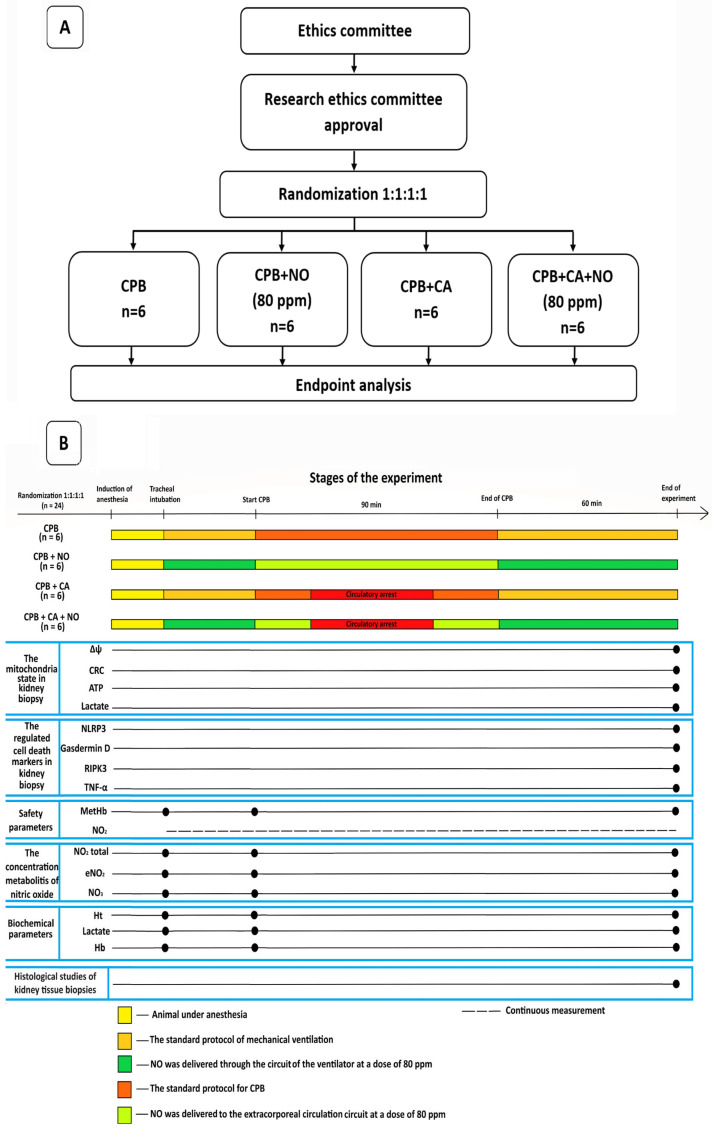
(**A**) A flowchart of the study design. (**B**) A flow diagram illustrating the stages of the experiment.

**Figure 3 biomedicines-12-01298-f003:**
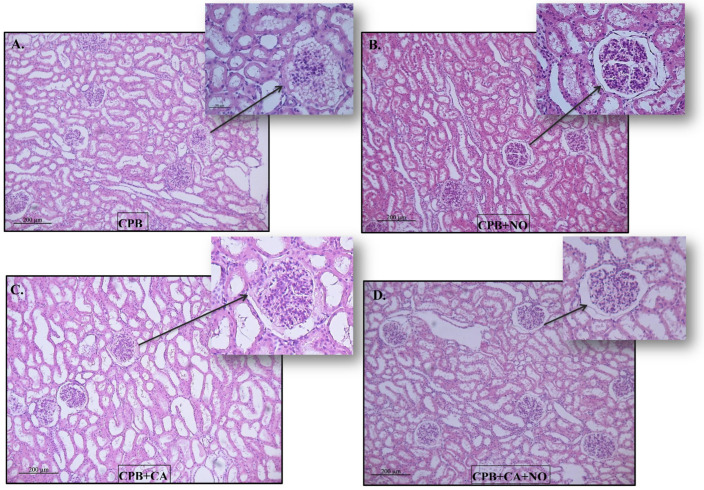
Histological pictures of kidney biopsy specimens 60 min after weaning the subjects from CPB: (**A**) CPB group, (**B**) CPB + NO group, (**C**) CPB + CA group, and (**D**) CPB + CA + NO group.

**Table 1 biomedicines-12-01298-t001:** Differences in main variables between CPB and CPB + NO groups at three stages of observation: tracheal intubation, CPB start, and 60 min after weaning from CPB; means ± SDs.

Variable	Group	Tracheal Intubation	CPB Start	60 min after Weaning from CPB	*p*-Value
Hb, g/L	CPB CPB + NO	95.50 ± 5.61 97.00 ± 4.10	94.50 ± 3.99 98.00 ± 1.79	83.83 ± 4.31 82.83 ± 4.71	0.00001 0.000002
Ht	CPB CPB + NO	29.0 ± 1.52 28.33 ± 2.42	27.67 ± 1.63 27.33 ± 1.75	24.83 ± 3.06 22.50 ± 2.95	0.0378 0.00016
MetHb %	CPB CPB + NO	0.45 ± 0.16 0.45 ± 0.30	0.48 ± 0.21 1.22 ± 0.21 *	0.58 ± 0.14 2.20 ± 0.34 *	0.194 0.00013
Urine output, mL	CPB CPB + NO	-	-	753 ± 191.34 972 ± 89.3	0.0297
uNGAL, ng/mL	CPB CPB + NO	-	-	1.69 ± 0.751 0.62 ± 0.268 *	0.0039
Lactate, mmol/L	CPB CPB + NO	1.70 ± 0.76 1.43 ± 0.53	1.66 ± 0.64 1.40 ± 0.41	3.61 ± 0.71 3.51 ± 0.61	0.0024 0.0002
e NO_2_, μmol/L	CPB CPB + NO	5.97 ± 0.671 6.92 ± 1.686	6.07 ± 0.624 6.89 ± 0.736	6.78 ± 1.486 7.31 ± 1.138	0.616 0.688
Total NO, μmol/L	CPB CPB + NO	24.21 ± 1.797 43.92 ± 15.281 *	28.21 ± 1.491 64.10 ± 14.733 *	25.09 ± 2.179 55.88 ± 9.177 *	0.158 0.023
NO_3_, μmol/L	CPB CPB + NO	18.21 ± 1.591 37.02 ± 14.435 *	22.21 ± 1.797 57.21 ± 14.885 *	18.376 ± 2.098 45.85 ± 9.402 *	0.286 0.013
NO treatment, minutes	189.67 ± 6.86 CPB + NO				
NO_2_ concentration, ppm	1.02 ± 0.17 CPB + NO				

NB: CPB—cardiopulmonary bypass; Hb—hemoglobin; Ht—hematocrit; NO—nitric oxide; NO_2_—nitrite (nitrogen dioxide); NO_3_—nitrate; uNGAL—Urinary Neutrophil Gelatinase-Associated Lipocalin. * indicates revealed statistically significant differences (*p* < 0.05) in mean values of variables in groups that underwent standard-protocol CPB and CPB + NO. *p*-values for urine output and uNGAL indicate intergroup differences; *p*-values for Hb, Ht, MetHb, Lactate, e NO_2_, Total NO, and NO_3_ indicate intragroup differences at experimental stages.

**Table 2 biomedicines-12-01298-t002:** Differences in main variables at three stages of observation in the CPB + CA and CPB + CA + NO groups: tracheal intubation, CPB start, and 60 min after weaning from CPB; mean ± SD values.

Variable	Group	Tracheal Intubation	CPB Start	60 min after Weaning from CPB	*p*-Value
Hb, g/L	CPB + CA CPB + CA + NO	95.83 ± 5.64 93.00 ± 5.76	96.83 ± 5.81 91.83 ± 4.40	82.0 ± 5.48 76.67 ± 3.83	0.00004 0.0007
Ht	CPB + CA CPB + CA + NO	29.00 ± 2.61 28.00 ± 1.41	28.33 ± 2.42 27.50 ± 1.87	22.67 ± 2.58 20.83 ± 2.64	0.00015 0.0001
MetHb, %	CPB + CA CPB + CA + NO	0.52 ± 0.16 0.57 ± 0.33	0.58 ± 0.26 1.28 ± 0.19 *	0.70 ± 0.23 2.28 ± 0.34 *	0.218 0.00025
Urine output, mL	CPB + CA CPB + CA + NO	-	-	678 ± 199.2 683 ± 201.9	0.77
uNGAL, ng/mL	CPB + CA CPB + CA + NO	-	-	2.23 ± 0.881 0.67 ± 0.255 *	0.0001
Lactate, mmol/L	CPB + CA CPB + CA + NO	1.47 ± 0.58 1.62 ± 0.64	1.37 ± 0.43 1.20 ± 0.35	4.72 ± 0.73 4.98 ± 0.37	0.0006 0.0003
e NO_2_, μmol/L	CPB + CA CPB + CA + NO	6.24 ± 0.776 6.20 ± 0.631	5.61 ± 0.405 5.94 ± 0.909	5.95 ± 0.489 6.61 ± 0.482	0.167 0.562
Total NO, μmol/L	CPB + CA CPB + CA + NO	26.02 ± 1.161 31.54 ± 3.560 *	26.41 ± 2.008 35.17 ± 3.036 *	26.08 ± 2.796 35.26 ± 3.951 *	0.101 0.513
NO_3_, μmol/L	CPB + CA CPB + CA + NO	19.78 ± 1.59 25.34 ± 4.010 *	20.80 ± 1.591 29.23 ± 3.117 *	18.29 ± 4.495 28.65 ± 4.171*	0.626 0.265
NO-treatment, minutes	190.17 ± 6.27 CPB + CA + NO				
NO_2_ concentration, ppm	1.22 ± 0.19 CPB + CA + NO				

NB: CA—circulatory arrest; CPB—cardiopulmonary bypass; Hb—hemoglobin; Ht—hematocrit; NO—nitric oxide; NO_2_—nitrite (nitrogen dioxide); NO_3_—nitrate; uNGAL—Urinary Neutrophil Gelatinase-Associated Lipocalin. * indicates revealed statistically significant differences (*p* < 0.05) in mean values of variables in groups that underwent standard-protocol СPB + CA and СPB + CA + NO. *p*-values for urine output and uNGAL indicate the intergroup differences; *p*-values for Hb, Ht, MetHb, Lactate, e NO_2_, Total NO, and NO_3_ indicate intragroup differences at experimental stages.

**Table 3 biomedicines-12-01298-t003:** The functional state of mitochondria in the CPB, CPB + NO CPB + CA, and CPB + CA + NO groups 60 min after weaning from CPB; mean ± SD values.

Variable	Group	60 min after Weaning from CPB	*p*-Value
Δψ, Ed/mg	CPB CPB + NO	126.7 ± 18.61 171.7 ± 20.41	<0.01 (CPB vs. CPB + NO) 0.24 (CPB + CA vs. CPB + CA + NO)
	CPB + CA CPB + CA + NO	108.3 ± 23.16 123.3 ± 18.61	0.26 (CPB vs. CPB + CA) <0.01 (CPB + NO vs. CPB + CA + NO)
CRC, Hm CaCl_2_/mg protein	CPB CPB + NO CPB + CA CPB + CA + NO	866.7 ± 216.02 1466.7 ± 02 366.7 ± 163.29 516.7 ± 147.19	<0.001 (CPB vs. CPB + NO) 0.21 (CPB + CA vs. CPB + CA + NO) 0.001 (CPB vs. CPB + CA) <0.001 (CPB + NO vs. CPB + CA + NO)
ATP, nmol/g	CPB CPB + NO CPB + CA CPB + CA + NO	3.7 ± 0.62 6.8 ± 1.1 3.95 ± 0.64 3.6 ± 0.72	<0.001 (CPB vs. CPB + NO) 0.39 (CPB + CA vs. CPB + CA + NO) 0.48 (CPB vs. CPB + CA) <0.001 (CPB + NO vs. CPB + CA + NO)
Lactate, mmol/L	CPB CPB + NO CPB + CA CPB + CA + NO	12.9 ± 3.71 10.2 ± 2.14 15.4 ± 2.6 12.2 ± 4.02	0.15 (CPB vs. CPB + NO) 0.14 (CPB + CA vs. CPB + CA + NO) 0.22 (CPB vs. CPB + CA) 0.22 (CPB + NO vs. CPB + CA + NO)

NB: ATP—adenosine triphosphate, CA—circulatory arrest, CPB—cardiopulmonary bypass, CRC—calcium retention capacity, NO—nitric oxide, Δψ—mitochondrial transmembrane potential.

**Table 4 biomedicines-12-01298-t004:** The data on regulated cell death in the CPB, CPB + NO CPB + CA, and CPB + CA + NO groups 60 min after weaning the subjects from CPB; mean ± SD values.

Variable	Group	60 min after Weaning from CPB	*p*-Value
TNF-α, ng/g	CPB CPB + NO	1349.5 ± 588.08 1538.9 ± 797.13	0.64 (CPB vs. CPB + NO) 0.03 (CPB + CA vs. CPB + CA + NO)
	CPB + CA CPB + CA + NO	1731.7 ± 497.27 1013.5 ± 422.07	0.26 (CPB vs. CPB + CA) 0.41 (CPB + NO vs. CPB + CA + NO)
RIPK3, ng/g	CPB CPB + NO CPB + CA CPB + CA + NO	10.2 ± 7.35 8.4 ± 3.86 15.1 ± 5.81 11.84 ± 5.01	0.59 (CPB vs. CPB + NO) 0.21 (CPB + CA vs. CPB + CA + NO) 0.23 (CPB vs. CPB + CA) 0.21 (CPB + NO vs. CPB + CA + NO)
NLRP3, ng/g	CPB CPB + NO CPB + CA CPB + CA + NO	208.8 ± 78.78 201.9 ± 38.96 199.6 ± 116.96 159.9 ± 82.49	0.85 (CPB vs. CPB + NO) 0.51 (CPB + CA vs. CPB + CA + NO) 0.68 (CPB vs. CPB + CA) 0.21 (CPB + NO vs. CPB + CA + NO)
GSDMD, ng/g	CPB CPB + NO CPB + CA CPB + CA + NO	27.5 ± 9.22 28.3 ± 11.64 29.4 ± 5.41 33.1 ± 13.76	0.88 (CPB vs. CPB + NO) 0.55 (CPB + CA vs. CPB + CA + NO) 0.72 (CPB vs. CPB + CA) 0.42 (CPB + NO vs. CPB + CA + NO)

NB: CA—circulatory arrest; CPB—cardiopulmonary bypass; GSDMD—gasdermin D; NLRP3—nucleotide-binding domain, leucine-rich–containing family, pyrin domain–containing-3; NO—nitric oxide; RIPK3—receptor-interacting protein kinase 3; TNF-α—tumor necrosis factor-α.

**Table 5 biomedicines-12-01298-t005:** Morphometric data on kidney injury in the CPB, CPB + NO CPB + CA, and CPB + CA + NO groups 60 min after weaning the subjects from CPB; mean ± SD values.

Variable	Group	60 min after Weaning from CPB	*p*-Value
GCI	CPB CPB + NO	2.09878 ± 0.9669 2.569 ± 0.6573	0.03 (CPB vs. CPB + NO) <0.0001 (CPB + CA vs. CPB + CA + NO)
	CPB + CA CPB + CA + NO	1.377 ± 0.3554 2.201 ± 0.7298	<0.001 (CPB vs. CPB + CA) 0.049 (CPB + NO vs. CPB + CA + NO)
LEI	CPB CPB + NO CPB + CA CPB + CA + NO	0.557 ± 0.3134 0.446 ± 0.1972 0.743 ± 0.2414 0.724 ± 0.3842	<0.01 (CPB vs. CPB + NO) 0.7 (CPB + CA vs. CPB + CA + NO) <0.0001 (CPB vs. CPB + CA) <0.0001 (CPB + NO vs. CPB + CA + NO)

NB: CA—circulatory arrest; CPB—cardiopulmonary bypass; GCI—glomerular–capsular index; LEI—lumen-epithelial index; NO—nitric oxide.

## Data Availability

Data will be made available upon request.

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
