# Peer review of "The Nephroprotective Effect of Nitric Oxide during Extracorporeal Circulation: An Experimental Study"

_biomedicines, 2024, doi:10.3390/biomedicines12061298_

Round 1

Reviewer 1 Report

Comments and Suggestions for Authors

This experimental study in sheep is an extension of previous clinical trials by this group showing cardiac and renal protection by inhaled NO during cardiac bypass operations, looking at renal morphology and possible mechanisms of renoprotection 1h after the intervention. With all that work, it is a pity GFR has not been derermined and that there is no data about renal function and morphology at later stages. 

Introduction, first sentence – it is suggested to provide data regarding the percentage of patients requiring RRT following bypass operations.

Please address the question: As to quoted references, such as 4-7, aside from KDIGO criteria for AKI, is there any evidence of NO reducing the need for RRT?

You may quote studies indicating the importance of intact intrinsic NO synthesis in preserving renal microcirculation and morphology, with an emphasize on medullary blood flow and parenchyma (Brezis, J Clin Invest 1991 Aug;88(2):390-5; Goldfarb, Kidney Int. 2001 Aug;60(2):607-13). This bears relevance to the current studies as NGAL likely reflects outer medullary hypoxic injury, principally affecting medullary thick ascending limbs (Heyman, Kidney Int. 2010 Jan;77(1):9-16).

In these perspectives, the study explores the tissue-protective impact of NO at the cellular level, but it is important to emphasize possible additional renoprotective mechanisms, such as the preservation of microcirculation and the control and attenuation of tubular transport under hypoxic conditions, as well as an impact of the cardioprotective effects on renal outcomes. In complementing scheduled studies, it is suggested to include evaluation of renal blood flows, and in particular medullary microcirculation

Methods: Please address the following questions: Where sheep in the CPB group maintained normothermic? Was renal blood flow maintained or held by cross-clamping of the aorta during surgery? If maintained – at what perfusion pressure? In other words, please make it clear if are we looking at cold/warm-ischemia-reperfusion injury or not?

Morphology – the described findings are non-specific and may indicate damage downstream to glomeruli and proximal tubules (glomerular and tubular dilatation). Did you evaluate medullary thick limbs? If it is a non-ischemia-reflow model, it might be interesting and physiologically important to look at medullary thick limb infrastructure, specifically mitochondrial swelling and nuclear pyknosis, that may appear within the time frame of the experiment.    

Tables 1,2 – I assume the p value on the right addresses within-group changes. Please make it clear.

Table 1 is redundant. On the other hand, I suggest adding a figure illustrating the differences in uNGAL and in urine output, illustrating to large extent the text addressing kidney protection (lines 352-358).

Since there is a difference in urine volumes between NO – treated and non-treated sheep, uNGAL should better be presented, corrected for uCreatinine.

Probably increased urine volume in NO-treated animals reflects decreased sodium retention. It might be interesting to assess sodium excretion    

Lines 462-7, how would you explain the different response to NO between CBP and CBP+CA protocols regarding mitochondrial function?

Comments on the Quality of English Language

Requires moderate improvement

Author Response

Dear Reviewer,

We are grateful to you for your work and great contribution to improving our manuscript. Your comments were so helpful and valuable.

Reviewer 2 Report

Comments and Suggestions for Authors

This is an interesting and important manuscript devoted to experimental results describing the nephroprotective effect of NO due to extracorporeal circulation. Because the temperature control of animals is important during general anesthesia and can influence postsurgical outcomes and experimental results, the authors should describe the maintenance and control of the intraexperimental animal body temperature in more technical detail. It should also be discussed how hypothermic and normothermic group results can be influenced by different temperature conditions of the animal body.

Typos (for example "...Alla A. Boshchenko Olga N. Serebryakova..." comma is missing) should be corrected.

Author Response

(The authors gave the same response as above.)

Reviewer 3 Report

Comments and Suggestions for Authors

Thank you for permitting me to review this manuscript 

In this study the authors studied the effect of NO after CPB in sheeps in four different scenario CPB+NO , CPB +CA +NO, and 2 shams groups

they suggest that NO administration lowered mitochondrial dysfunction

Introduction 

Line 71 after declaring the objective adequately  the authors cite that understanding the protective effect of NO is important since this last sentence (protective effect of NO) is not verified , this should adjusted to circumstances 

results 

Table 1 should be readjusted , why some lines are with one group and others with 2 groups , were are the sham groups?

Since the main conclusion is based on results which are not statisticlly significant , the summary of the finfings and conclusion need to me reshaped and rewritten in order to commumicate the facts found in this study only

Author Response

(The authors gave the same response as above.)

Round 2

Reviewer 1 Report

Comments and Suggestions for Authors

Lines 99, 102, 104 – are these perfusion pressures(??) or rather flows, expressed al L/min/1/73m2?

I suggest that you add at the bottom of the Discussion a brief list of limitations of the study, starting by the short post-op observation period, the lack of renal functional indices other than urine output, and that lack of detailed renal outer medullary morphology, a region of supreme importance in hypoxic AKI (with the current reference number 39).

You should also clearly state that currently there is no evidence that NO reduces the need for RRT following bypass operations, and that this should be explored in further more extended clinical and laboratory studies

I think you should also provide a short explanation for the different outcomes of the two models, stating that it may reflect different degrees of hypoxic stress and lethal/sublethal injuries

Please note that CsA nephropathy is in essence mediated by hypoxia, attributed principally to profound vasoconstriction (Fahling, Acta Physiologica (Oxf) 219:625-639, 2017). Stating that may provide the ignorant reader with the rationale of addressing this type of injury in your Discussion      

Comments on the Quality of English Language

none

Author Response

Dear Reviewer,

We again would like to express our gratitude for your careful analysis of our work that significantly contributes to its improvement.
